# Predictors of Inadequate Health Literacy among Patients with Type 2 Diabetes Mellitus: Assessment with Different Self-Reported Instruments

**DOI:** 10.3390/ijerph20065190

**Published:** 2023-03-15

**Authors:** Marija Levic, Natasa Bogavac-Stanojevic, Dragana Lakic, Dusanka Krajnovic

**Affiliations:** 1PhD Program of Department of Social Pharmacy and Pharmaceutical Legislation, Faculty of Pharmacy, University of Belgrade, 11221 Belgrade, Serbia; 2Department of Medical Biochemistry, Faculty of Pharmacy, University of Belgrade, 11221 Belgrade, Serbia; 3Department of Social Pharmacy and Pharmaceutical Legislation, Faculty of Pharmacy, University of Belgrade, 11221 Belgrade, Serbia

**Keywords:** functional health literacy, critical health literacy, patients, diabetes mellitus type 2, self-reported instruments, multivariate logistic regression

## Abstract

Introduction: Obtaining, understanding, interpreting, and acting on health information enables people with diabetes to engage and make health decisions in various contexts. Hence, inadequate health literacy (HL) could pose a problem in making self-care decisions and in self-management for diabetes. By applying multidimensional instruments to assess HL, it is possible to differentiate domains of functional, communicative, and critical HL. Objectives: Primarily, this study aimed to measure the prevalence of inadequate HL among type 2 diabetes mellitus patients and to analyze the predictors influencing health literacy levels. Secondly, we analyzed if different self-reported measures, unidimensional instruments (Brief Health Literacy instruments (BRIEF-4 and abbreviated version BRIEF-3), and multidimensional instruments (Functional, Communicative and Critical health literacy instrument (FCCHL)) have the same findings. Methods: The cross-sectional study was conducted within one primary care institution in Serbia between March and September 2021. Data were collected through Serbian versions of BRIEF-4, BRIEF-3, and FCCHL-SR12. A chi-square test, Fisher’s exact test, and simple logistic regression were used to measure the association between the associated factors and health literacy level. Multivariate analyses were performed with significant predictors from univariate analyses. Results: Overall, 350 patients participated in the study. They were primarily males (55.4%) and had a mean age of 61.5 years (SD = 10.5), ranging from 31 to 82 years. The prevalence of inadequate HL was estimated to be 42.2% (FCCHL-SR12), 36.9% (BRIEF-3) and 33.8% (BRIEF-4). There are variations in the assessment of marginal and adequate HL by different instruments. The highest association was shown between BRIEF-3 and total FCCHL-SR12 score (0.204, *p* < 0.01). The total FCCHL-SR12 score correlates better with the abbreviated BRIEF instrument (BRIEF-3) than with BRIEF-4 (0.190, *p* < 0.01). All instruments indicated the highest levels for the communicative HL domain and the lowest for the functional HL domain with significant difference in functional HL between the functional HL of FCCHL-SR12 and both BRIEF-3 and BRIEF-4 (*p* = 0.006 and 0.008, respectively). Depending on applied instruments, we identified several variables (sociodemographic, access to health-related information, empowerment-related indicators, type of therapy, and frequency of drug administration) that could significantly predict inadequate HL. Probability of inadequate HL increased with older age, fewer children, lower education level, and higher consumption of alcohol. Only high education was associated with a lower probability of inadequate HL for all three instruments. Conclusions: The results we obtained indicate that patients in our study may have been more functionally illiterate, but differences between functional level could be observed if assessed by unidimensional and multidimensional instruments. The proportion of patients with inadequate HL is approximately similar as assessed by all three instruments. According to the association between HL and educational level in DMT2 patients we should investigate methods of further improvement.

## 1. Introduction

The World Health Organization (WHO) and the International Diabetes Federation estimated that in 2019, 463 million people worldwide suffered from diabetes mellitus (DM) and that the number of DM patients will increase to 700 million by 2045. Although the highest incidence rates are registered in developed countries, a significant increase in the number of patients is expected in developing countries, including Serbia [1]. According to an estimate by the Institute for Public Health of Serbia, approximately 770,000 people—12.0% of the adult population—suffer from DM in the Republic of Serbia [2]. According to the estimates of domestic experts and based on the results of international studies, 43% (330,000) of patients with type 2 DM (DMT2) have not been diagnosed and are not aware of their disease [3,4,5]. The number of people with DM is many times higher (95%) compared to people with type 1 DM (DMT1). In Serbia, as well as in the world’s developed countries, DM is the fifth leading cause of mortality [6] and the fifth-highest cause of disease burden [7].

The main goal of today’s health system is to promote and maintain good health while enabling people to take care of their health and to participate more in making decisions related to their health [8,9,10]. Participation in health decisions could be moderated by specific patients’ skills and by the situational demands and complexities experienced by patients in their attempts to obtain, understand, and use health information or health services. As a social determinant of health, HL is a crucial driver to approaching health information competently and effectively, making self-care decisions, and taking self-management actions for diabetes. Recognizing that individual responses to information will result in different learning outcomes and associated behavioral and health outcomes led us to research that focuses on the categorization of functional health literacy (FHL), communicative/interactive health literacy (IHL), and critical health literacy (CHL). FHL is often required to meet the immediate and necessary goals of clinical care as it is related to the specific skills required to achieve outcomes that are determined primarily by those providing healthcare [11]. In such circumstances, specific skills to manage prescribed activities in chronic therapies could be required of any patient with chronic diseases at a point of decision-making. In contrast, IHL and CHL require the development of transferable skills, including obtaining, understanding, evaluating (interpreting), and acting on (applying) health information, which enable patients to engage with and make health decisions in various contexts. The development of specific and transferable skills offers a greater opportunity to optimize HL’s contribution in mediating self-care, low adherence, and medication management for a chronic disease such as diabetes. Several studies found that a small number of DMT2 patients had adequate levels of HL [12,13,14,15,16,17]. Some studies showed a connection between low-level HL, limited knowledge about the disease [18,19], and poor glycemic control [19,20,21]. Diabetic individuals with lower HL use preventive health services less, are more at risk of misdiagnosis, experience difficulties managing chronic diseases, have poor drug and treatment compliance, and have poorer health outcomes [15,17].

As self-care for a chronic disease such as diabetes often relies on information given in verbal instructions, printed educational materials, and patient education courses [12], low-literacy patients may have problems finding and following these instructions when they are to be integrated into everyday life. Inadequate HL (InHL) has been associated with poorer health states, broader inequalities, and higher health system costs. Hence, measuring changes to the specific skills required by DMT2 patients for decision-making and the more generic transferable skills that enable well-informed and more autonomous health decision-making could be considered crucial for diabetes self-management [12,21].

Based on the previous considerations, different self-reported and objective measures are needed if we want to assess both FHL as well as IHL and CHL. It is already proven that instruments vary in how they operationalize the concept of HL into a measurable construct, and many address limited sets of conceptual dimensions of HL. Measures that have the broadest measurement scopes are considered the most suitable for application in diabetes reference. In this research, three instruments were used: the Functional, Communicative, Critical, Health Literacy instrument (FCCHL) and the Brief Health Literacy Screening Instruments with three questions (BRIEF with three (BRIEF-3) and four questions (BRIEF-4)). The use of the BRIEF instruments in addition to FCCHL would serve as an additional confirmation of the distribution of diabetics in the category of those with high (adequate/AHL) or low (inadequate/InHL) HL.

Keeping in mind that WHO claims HL is one of the most critical determinants of health [21], this study aims to assess HL and its domains among DMT2 patients in Serbia and identify predictors of InHL. Furthermore, the authors analyzed if different self-reported measures, screening unidimensional (BRIEF-3 and BRIEF-4 instruments), and multidimensional (FCCHL-SR12) have the same findings.

## 2. Method

A cross-sectional study was conducted from March to September 2021 using non- probability sampling within one healthcare center in the municipality of Belgrade. A convenient sample of primary care patients with DMT2 was used. Before the survey, authors recruited five research assistants. To ensure that they were familiar with the purpose, process, and procedure of applying the instrument, the authors systematically trained three pharmacy graduates and two doctors as research assistants. Throughout data collection, the interviewers (researchers and assistants) explained the purpose and significance of the study to the participants and obtained written informed consent. The survey was conducted in a doctor’s office after the completed examinations. Patients filled out the questionnaires themselves (self-administrated measurements), and interviewers were at their disposal during this time. They did not receive any payment for filling out the instrument. All data were anonymous and were entered into the database as such.

The required number of participants was calculated based on the population of DM patients in the Belgrade (80,241) area. An estimated percentage of high HL from the literature (which was 36% [22,23,24]) was used for the calculation, and 95% confidence interval with an error of 5% were used for calculation. Based on these parameters, the required sample was at least 353. The required sample size of 353 was increased by 10% due to potential dropouts (accounting for the non-responder rate) during the study.

### 2.1. Sample and Data Collection

The target population was patients diagnosed with DMT2 at least six months before the start of the study who knew the Serbian language, were over 18 years old, and voluntarily agreed to participate with signed informed consent. Exclusion criteria were medical background (such as doctors, nurses, pharmacists). Three instruments were used, which participants filled out anonymously and voluntarily after receiving comprehensive information from the interviewer). The interviewers first explained the aim and the course of the research; those patients who wanted to participate filled in the questionnaires, and one medical parameter, HbA1c value, was taken from the medical documentation on the participants’ agreement.

### 2.2. Instruments

Participants were given three instruments at the same time to complete in the following order: general instrument, instruments for rapid screening of health literacy (BRIEF-3 and BRIEF-4), the 12-item Serbian version of the three-dimensional instrument for the assessment of health literacy (FCCHL-SR12).

The general instrument is a specially created instrument that includes questions related to the social and demographic characteristics of the participants, health-related issues, health behaviors, as well as empowerment-related indicators. The instruments’ questions were generated using a combination of literature review methods and analysis of instruments used in similar research conducted in diabetic populations. In line with the Sørensen model [22], many outcomes associated with HL were collected: self-assessed health status (excellent, very good, good, so-so/fair, bad), duration of diabetes, type of therapy for diabetes, and frequency of therapy. To measure health status, participants were asked about other long-term illnesses (illnesses that have lasted or are expected to last for at least 6 months), HbA1c value, coded in ≤7% and >7%, and do not know/refuse. In addition to including questions on gender, birth year, educational level, and marital status, the questionnaire also has questions on the number of family members living in the same household, employment status (currently having a paid job), financial situation, and the number of children. Health behaviors (measured using three items on physical activity, tobacco use, and alcohol consumption), access to health-related information (a primary source of information), and empowerment-related indicators (perceived interest in one’s health and perceived self-assessment of one’s health in general) were recorded as well [23,24].

FCCHL is a multidimensional instrument that can be considered the most valuable and comprehensive instrument in screening for InHL [25]. BRIEF instruments focus on function, but contain the interaction, comprehension, and self-efficacy dimensions of HL. BRIEF instruments are primarily used for quick assessment, and they have many advantages in comparison to other instruments, including that they are less likely to cause anxiety and shame [22,25,26,27,28,29]. All three instruments can be considered the best for measuring FHL in patients with DMT2 and other chronic diseases [23,24]. They are self-reported instruments and are quick, easy, and inexpensive to administer. The authors used culturally adapted versions of these measures developed in previous studies for the Serbian population that are valid and reliable.

BRIEF-3 consists of three questions and BRIEF-4 of four, which are evaluated on a five-point scale of 1–5. Both instruments can be administered and scored in less than two minutes. Each item is worth 1 to 5 points depending on their response, and summated responses provide a total score from a minimum of 3 to a maximum of 15 in BRIEF-3 and from a minimum of 4 to a maximum of 20 in BRIEF-4. We used the same scoring system as other researchers [28,29,30], categorizing individuals’ functional health literacy as inadequate HL/InHL (points 3–9), marginal HL/MHL (points 10–12), and adequate HL/AHL (points 13–15) in BRIEF-3 and InHL (points 4–12), MHL (points 13–16) and AHL (17–20) in BRIEF-4.

The 12-item Serbian version of the three-dimensional FCCHL (FCCHL-SR12) is a multidimensional scale that uses subscales to measure different but related aspects to capture the complexity of HL that is context- and content-specific for the assessment of three dimensions of HL (functional, communicative, and critical) as well as total HL. It consists of 12 items and uses a 4-point Likert scale (1–4). The instrument contains questions that ask how often (never to often) patients have trouble with reading or understanding leaflets from healthcare providers/hospitals or pharmacies (FHL; 4 items) and how difficult it is (easy to rather difficult) to perform specific actions concerning health information (IHL (4 items) and CHL (4 items)). Since there are no defined cutoffs for InHL, MHL, and AHL for FCCHL-SR12, authors used defined levels according to the same principle as for BRIEF instruments. The level of HL measured using BRIEF was categorized as InHL (up to 60% of a maximum score), MHL (up to 80% of a maximum score), or AHL (more than 80% of maximum score). The levels of FHL, IHL, CHL, and total HL measured using the FCCHL-SR12 instrument were defined in the same way. The FCCHL-SR12 score range is 12–48, and the scoring is as follows: 12–28 (InHL), 29–38 (MHL), and 39–48 (AHL)

Investigating the dimensionality and validity of the FCCHL-SR12 was reported elsewhere [31], showing acceptable psychometric properties. A 3-dimensional 12-item version of the FCCHL had acceptable psychometric properties.

Confirmatory factor analysis provided a good statistical and conceptual fit for the data. The analysis of the internal consistency of the FCCHL-SR12 was satisfactory for the health literacy total scores (Cronbach’s alpha was 0.767) and also acceptable for the three correlated subscales (Cronbach’s alpha equaled 0.792, 0.748, and 0.796 for FHL, IHL, and CHL, respectively) [31].

### 2.3. Statistical Analysis

Data were managed and analyzed using IBM SPSS Statistics for Windows, version 27.0. Armonk, NY, USA: IBM Corp software package. The significance level was set at a 95% confidence level, with a *p* value of less than 0.05.

Descriptive statistics, such as frequencies (absolute and relative), were used to describe the sample. Mean and standard deviation (SD) were employed for continuous data (such as age). To determine the relationship between HL scales, Spearman’s correlation coefficient was calculated. The distributions of sociodemographic, health-related, and health behavior characteristics along with health-related information and empowerment-related indicators through different HL levels were compared via chi-square test. Univariate and multivariate logistic regression analysis were used to determine independent predictors of inadequate HL as evaluated by all three instruments.

### 2.4. Ethical Considerations

The study adhered to the ethical standards in line with the International Ethical Guidelines for Health-related Research Involving Humans (the Council for International Organizations of Medical Sciences, CIOMS, 2016) and Declaration of Helsinki (World Medical Association, 2013).

Approvals to conduct the research were obtained from the ethics committee of the primary healthcare institution in Belgrade.

## 3. Results

A total of 385 people were approached, of which about 90% agreed to participate. The final sample consisted of 350 DMT2 participants. Participants were predominantly male (55.4%) and had a mean age of 61.5 (SD = 10.5) years, ranging from 31 to 82 years. Participants’ data are presented in Table 1.

### Distributional Properties

The mean scores for each domain of the FCCHL-SR12, BRIEF-4, and BRIEF-3 instruments are presented in Table 2. Regarding the FCCHL-SR12 instrument, the highest levels were for the IHL domain, and the lowest was for the FHL domain. Higher levels were found for BRIEF-4 instruments in comparison to the BRIEF-3 instrument.

Items in all instruments showed no skewness or kurtosis in the distribution of scores. In BRIEF-4 and BRIEF-3 instruments, kurtosis was negative and indicated the small outliers in a distribution. The distributions of scores for FHL, IHL, CHL, FCCHL-SR12, BRIEF-4, and BRIEF-3 are presented in Table 2.

The relationship between the total FCCHL-SR12 score and those of BRIEF instruments was investigated. The weak correlation was shown for total FCCHL-SR12 with BRIEF-3 (r = 0.204, *p* < 0.01) and BRIEF-4 (r = 0.190, *p* < 0.01). Both BRIEF instruments measure FHL, so we evaluated the association between BRIEFs and FCCHL-SR12 FHL domain. Neither the BRIEF-4 nor the BRIEF-3 instrument demonstrated a good correlation with the FHL domain (r = 0.034, *p* = 0.526 and 0.037, *p* = 0.490, respectively). 

Figure 1 shows the distribution of InHL, MHL, and AHL as assessed by different instruments and the FHL domain of FCCHL-SR12. Concerning the level of knowledge measured using different instruments, there is a significant difference between FCCHL-SR12 and BRIEF-3 (*p* = 0.003), in contrast to FCCHL-SR12 and BRIEF-4, where no statistically significant difference was observed (*p* = 0.192). The proportion of patients with InHL is approximately similar, but there are variations in the assessment of MHL and AHL. The difference in AHL ranges from 3.6% for FCCHL-SR12 to 14.8% for BRIEF-3. Additionally, the difference in HL levels between FHL of FCCHL-SR12 and both BRIEF-3 and BRIEF-4 was significant (*p* = 0.006 and 0.008, respectively).

Since all three instruments similarly assess InHL, agreement in HL levels determined by BRIEFs and FCCHL-SR12 questionnaires was observed. The results are shown as a heat map for a cross table (Figure 2). Even though a similar number of participants have InHL for both instruments, 116 (33.3%) (FCCHL-SR12) vs. 118 (33.8%) (BRIEF-4) (Figure 1); only 49 of them were classified to have InHL with both instruments (Figure 2a). Using BRIEF-3 and FCCHL-SR12 instruments, only 55 participants were concurrently ranked in the InHL group (Figure 2b). BRIEF-3 and BRIEF-4 overestimate InHL in 12 patients and underestimate AHL in 4 patients evaluated by FCCHL-SR12.

Table 3 shows the distribution of HL levels concerning the participants’ sociodemographic characteristics. All instruments identified the dependence of InHL on education level, exercise, and alcohol consumption. InHL was more prevalent in less-educated patients, in those who exercised rarely, and those who often consumed alcohol. HL levels also depended on gender, number of children, employment status, and interest in health as determined by BRIEF instruments. InHL were the most prevalent in males, in participants with two or more children, unemployed and pensioners and in participants who were not interested in health. However, when FCCHL-SR12 was used, HL literacy levels depended on the type of therapy, frequency of drug administration, and smoking status. InHL was more prevalent in patients who used diet and drugs as a therapy, who less frequently administrated drugs, and who were smokers.

Furthermore, considering that all three instruments identify persons with HL in a similar percentage but the number of participants who were classified in the same way was small, the participants were divided into those with InHL and those with MHL and AHL to examine the predictors for InHL. Sociodemographic characteristics of participants (gender, marital status, children, education, employment, income, therapy, frequency of administration, health behaviors (exercise, alcohol, smoking), access to health-related information, and empowerment-related indicators (interest in health and self-estimation of health status)) were used as predictors of InHL. Predictors of InHL assessed by BRIEF-4 or BRIEF-3 were age, male gender, education level, employment status, and number of children. Probability of InHL increased with older age, number of children, and if participants were unemployed, contrary to female gender and high education, which decreased it. If InHL was assessed by BRIEF-3, additional predictors were exercise level and smoking status, reducing the probability of IHL. In the case of FCCHL-SR12—predictors were education level, smoking status, alcohol consumption, type of therapy, and frequency of drug administration. Data are shown in Table 4.

Additionally, all significant predictors were included in multivariate analysis to assess independent predictors of InHL. Education was a significant independent predictor of InHL level for all three instruments. High education was associated with a lower probability of InHL. If the independent predictors of BRIEF-3 and BRIEF-4 are compared, it can be seen that the common predictors (except education) were age. Additionally, the number of children is an independent predictor for BRIEF-3. A higher number of children and older age were associated with a higher probability of InHL. Alcohol was an independent predictor for FCCHL-SR12. Lower consumption of alcohol is associated with a lower probability of InHL levels. 

## 4. Discussion

### 4.1. Health Literacy Levels

Results showed that using the FCCHL-SR12 instrument among the DMT2 patients in Serbia, HL scored the best on IHL, followed by CHL and FHL, providing support for Nutbeam’s model on the three types of HL and levels of complexity. Patients in Serbia may be more functionally illiterate, but given that they have to learn to manage their illness, they are better at communication literacy. These results were not in accordance with [17,32,33], who found CHL to be the most difficult. As an explanation as to why CHL was not the most difficult among the participants in Serbia is that CHL comprises more advanced cognitive skills, but chronic patients may have developed self-management skills to manage the therapy and the disease. Perhaps, some issues are already defined in DM because there are treatment protocols, etc., have an influence. Considering that all patients who met the criteria were offered an opportunity to participate, even those with low literacy (e.g., FHL, who would have refused if it was electronic or post-survey), they had the same chance to be included in this type of study design. This is the explanation as to why in the sample we have many patients with low FHL. However, it is obvious that these people are either well cared for by their caregivers or that the patients are well trained to manage the disease and have developed IHL and CHL. In addition, the FCCHL scale is validated for a specific group of patients in Serbia—for diabetics—so the question is whether the same results would be obtained when measuring HL in the general population.

With the use all three instruments, the proportion of patients with InHL was similar (FCCHL-SR12 detects HL similarly to BRIEF-3 and in some percentages more than BRIEF-4), but there are variations in the assessment of AHL (it ranges from 3.6% for FCCHL-SR12 to 14.8% for BRIEF-3) and MHL. In addition, a very weak correlation was shown between FCCHL-SR12 and both BRIEFs. FCCHL-SR12 detects a very small percentage of adequate HL. A very weak correlation between BRIEF-3 and FCCHL-SR12 was confirmed by the poor agreement of the instruments. Unlike other studies investigating FCCHL (17, 32, 33), higher correlation was found between IHL and CHL than between FHL and those two subscales. In this research, most patients had MHL (63.3%, 53.0%, and 48.3% measured using FCCHL-SR12, BRIEF-4, and BRIEF-3, respectively) with a broader range of AHL (from 3.4% for FCCHL-SR12 to 14.83% for BRIEF-3). Some previous studies that evaluated HL levels measured using different instruments in the same population showed that a low proportion of patients had AHL levels, with a reported prevalence ranging from 15% to 40%. Many of these studies were performed in developed western countries (the USA and the UK) [29,34,35]. However, there has been limited explanation of the observed differences in the prevalence, and there was no effort to look at this problem globally. The proportion of patients with InHL in our study was from 33% to 37%, and it was similar to some studies conducted in the US (32.8%, 26.3%, 37.2%), Brazil (26.7%), and the Marshall Islands (24%) [36]. The study with the highest prevalence of InHL (82%) was conducted in 2012 in Taiwan [37], and the lowest prevalence of InHL (7.3%) was reported in 2011 in Switzerland [38] among DMT2 patients. It is found that the there is a need for countries to measure the burden of InHL in DMT2 patients using one standardized tool. A standardized method of measuring HL would allow for a direct comparison of findings between countries [39].

### 4.2. Patients Characteristics and Predictors for InHL

After investigation of HL across levels of social and demographic characteristics of the participants, health-related issues, health behaviors, and empowerment-related indicators, it is shown that DMT2 patients with higher education, males, the employed, and patients interested in their health had significantly higher HL than their counterparts. This finding was valid when applying the FCCHL-SR12, BRIEF-3, and BRIEF-4 instruments. The absence of an association between HL and gender was found in the research from Finbråten et al. [40], while in research from Hussein et al. [41], InHL was more likely to be higher in females.

A significant difference was found in HL with regards to age with the BRIEF-3 and BRIEF-4 instruments. These results are in accordance with Al Sayah et al. [42], van der Heide et al. [19], and Hussein et al. [41]. Hence, higher age indicates lower HL level, which is in contrast with Al Sayah et al. [25] and Vandenbosch et al. [43] and in line with the findings from Heijmans et al. [17]. Furthermore, with the BRIEF-3 and BRIEF-4 instruments, age is shown as an independent predictor. The reasons for not detecting those differences by FCCHL-SR 12 might be related to the multidimensionality of the instrument rather than the measurement scope.

Regardless of the instruments used, significant differences in HL were found in relation to education. People with DMT2 who had completed a university-level education reported a significantly higher HL than those with secondary school and lower educational levels. This is in line with the findings from Heijmans et al. [17], van der Heide et al. [19], Hussein et al. [41], Vandenbosch et al. [4], Berkman et al. [44], and Jeppesen et al. [45], who used different instruments; and those that used FCCHL to measure HL in DMT2 population were Nutbeam [11], Al Sayah et al. [25], and Finbråten et al. [40]. However, the average age of the sample was relatively high, so differences related to age may not have been evident. In the research conducted by Abdullah et al. in Malaysia, there was no significant association between educational level and HL [46].

Higher ages and lower education are in direct correlation with lower capacity of people to make sound decisions in the context of their everyday life; their ability to protect, maintain, and increase control over their illness and health is diminished. Poor health and worse health outcomes are consistently found among patients with more complex care need; these findings highlight the potential role of HL in this relationship. Differing from other research from Jeppesen et al. [45] and Finbråten et al. [40], an association between HL and health behaviors (alcohol consumption and smoking habits) was found with FCCHL-SR12 and confirmed by the combination of BRIEF-3 and FCCHL-SR12. Recommendations about smoking and alcohol consumption have been promoted among people with diabetes, and therefore, information on smoking and alcohol risk might be easier to understand regardless of HL level compared to other health behaviors.

Number of children is an independent predictor for the abbreviated BRIEF, and it is associated with a higher probability of inadequate FHL, as parents with two or more children reported lower levels of FHL. The study conducted among the parents of preschool children in Serbia reported higher total pharmacotherapy scores (PTHL) among higher families using the PTHL-SR instrument [47]. In the research conducted in 2011, the number of children was associated with a lower probability for InHL [48]. The reasons for these differences are not clear and point out the need for future research. It may be that different results depending on yet-to-be elucidated factors, such as other parental characteristics. In chronic diseases such as diabetes, in which the patients should provide their own diabetes management (compliance with medication regimen, diet, blood glucose level measurement, insulin administration, foot care, etc.), the individual must be informed about HbA1c and its value as one of the most important metabolic markers in diabetes. The value of HbA1c being above the targeted value shows insufficient compliance with the treatment and care. We found InHL among patients with HbA1c>7, assessed using FCCHL-SR12. InHL leads to poorer diabetes self-care management skills, which may affect the control of HbA1c levels. To ensure sufficient diabetes self-care management skills, it is necessary for patients with diabetes to possess a high level of HL. These skills are needed in day-to-day decision-making, such as when measuring blood sugar levels, and patients with diabetes need to respond with the appropriate action for the reading they receive. Healthcare providers can improve the self-care management skills of patients with diabetes by enhancing their health literacy level through educational means, both face-to-face and through written information. This finding addresses the importance of not only treating the individual’s disease but also assessing and strengthening their health literacy level.

Al Sayah et al. [42], Finbråten et al. [40], and Lee et al. [48] did not find any significant difference in HL related to HbA1c using the same FCCHL instrument. However, Finbråten et al. found significant differences in HL between those patients reporting and those not reporting their latest HbA1c levels [40]. Some previous studies reported inconsistent results of possible association between HL and HbA1C, investigated by Bailey et al. [12], Haun et al. [29], Franzen et al. [38], Al Sayah et al. [42], Jeppesen et al. [45], and Bains et al. [49]. The reason for that inconsistency might be in a fact that different instruments were used in those studies—mostly those that are limited to FHL.

With the FCCHL-SR12 instrument, DMT2 patients who use insulin had a higher level of HL in comparison with those on tablets, which is in line with an expectation that they have higher knowledge and are engaged more actively in therapy management.

### 4.3. Strengths and Weakness

In Serbia, studies on HL are insufficient, especially in the primary healthcare sector. To our knowledge, this is the first study that evaluates HL with instruments which are validated and have satisfactory validity and reliability, so that they can be used in population research.

The strength of the study is that it was the first study to assess patients’ health literacy in DMT2 in primary healthcare, which accounts for 80% of health-related decisions.

Therefore, the findings from this study did not only reveal levels of HL literacy among this patient population but also suggested recommendations for healthcare professionals.

Although the sample size was interpreted as sufficient for discussions and drawing sufficiently trustworthy conclusions, some limitations must be acknowledged. Data were collected using self-administrated measurements, which could be quite challenging for those people with InHL, especially those with low FHL, as the items require reading comprehension abilities. It is necessary to consider the possibility that those who chose to participate in this study had a higher HL. However, the recruitment approach and the pandemic measures could have assured the inclusion of individuals with very low HL levels. Patients who would refuse to participate in postal survey or online survey or who are illiterate or have problems with understanding the content of the instrument may have participated. This bias could have led to recruiting more individuals with low levels of HL, so the reported prevalence could be an overestimation of the true prevalence in this population. Due to the COVID pandemic and special measures issued on the doctors’ visits, some participants preferred that researchers filled out the survey instruments. In non-pandemic circumstances, those people might refuse to self-administer the instrument. In addition, the instruments were quite lengthy and could have been quite stressful and/or time-consuming to fill out, which also might be a reason for non-responses or drop-outs, if the research would have been done in non-pandemic circumstances. Still, during the validation process of the FCCHL-SR12, all participants (regardless of educational level) reported that the items were clearly stated.

The study was conducted at a single healthcare center at the capital of Belgrade, which could limit the generalization of the results. This is a typical primary healthcare institution for those patients, and it fully represented a demographically diverse population. The adequate age and gender distribution of the sample reflected the targeted population well and could be considered representative of elderly people of DMT2 in the country [50]. The mean age of people with DMT2 in this study was relatively high (61.5 years). In Serbia, the incidence of DMT2 is significantly higher in the elderly and in men. According to the Statistical Office of the Republic of Serbia, in 2021, observed by gender, 51.3% were women and 47.7% were men. At the same time, the process of demographic aging of the population is manifested as low participation of young people and a high and continuously growing share of the elderly in the total; according to the data, the share of people over 65 years of age is 21.3% [49]. However, the educational level in the sample was higher than that of the general Serbian population; according to the 2011 census in Serbia, 16.2% of inhabitants have higher education, 49% have a secondary education, 20.7% have an elementary education, and 13.7% have not completed elementary education [51].

Another limitation of the study is related to the cross-sectional design, making it impossible to discuss the role of causality in the associations found between predictors of HL levels and patients’ characteristics. Prospective study designs could more accurately describe the relationship between the two. Because the study did not use a randomized, controlled design, it may fail to account for confounding variables that introduced measurement errors. Additionally, three perception-based measures were included, and a performance-based measurement of HL was not, making interpretation challenging.

## 5. Conclusions

Beyond previous measures focusing only on FHL (BRIEFs instruments), FCCHL-SR12 includes three levels of HL, each of which might have different effects on patient outcomes. BRIEF instruments appeared to point in the same direction as FCCHL-SR12. Both BRIEF and FCCHL-SR12 can be easily administered in a healthcare setting. Among primary care DMT2 patients in Serbia, MHL was dominant, with a low proportion of patients with AHL. Our findings indicate that parental status with more children and greater frequency of alcohol use are predictors of InHL and confirm those of previous studies showing that InHL is also associated with lower educational level and higher ages. However, by applying all three instruments, we found only education as an independent predictor. Future investigation is necessary to support those findings on a larger population and to improve our understanding of factors that could help primary care practitioners in the future to more easily identify patients who need help with the management and application of information for better control of their disease.

## Figures and Tables

**Figure 1 ijerph-20-05190-f001:**
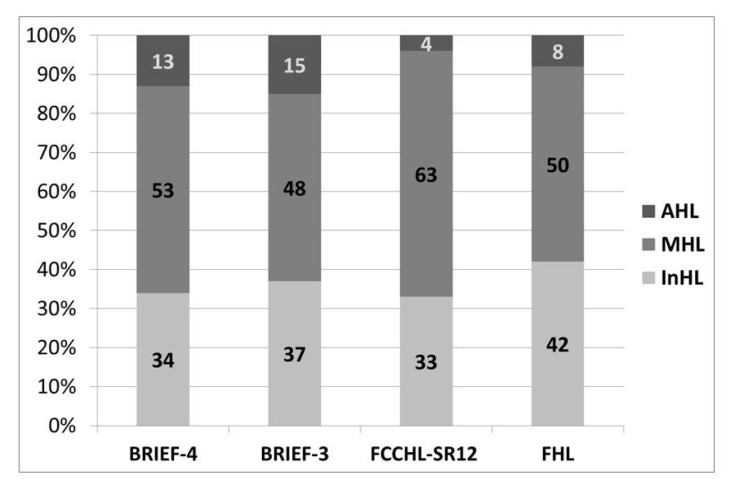
Distribution of HL levels among instruments. AHL—adequate health literacy, MHL—marginal health literacy, InHL—inadequate health literacy.

**Figure 2 ijerph-20-05190-f002:**
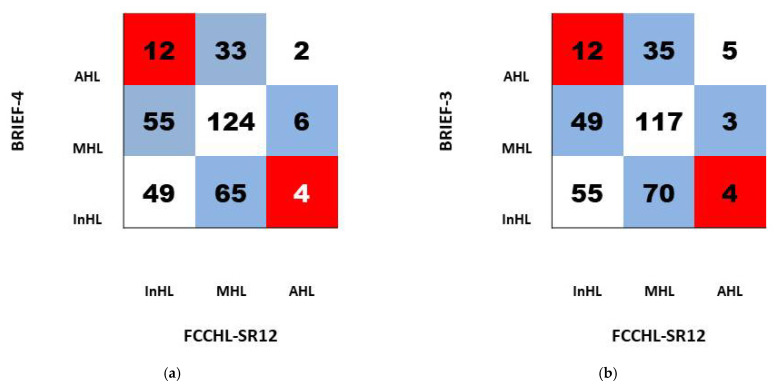
The white boxes show the number of respondents selected similarly with both FCCHL-SR12 and BRIEF-4 (**a**) or FCCHL-SR12 and BRIEF-3 (**b**) instruments. The blue boxes represent HL levels overestimated or underestimated by BRIEFs compared to FCCHL-SR12 for one level of knowledge. Red boxes represent overestimated or underestimated HL levels by BRIEFS for two levels of knowledge. AHL—adequate health literacy, MHL—marginal health literacy, InHL—inadequate health literacy.

**Table 1 ijerph-20-05190-t001:** Sociodemographic characteristics, health-related characteristics, health behaviors, access to health-related information, and empowerment-related indicators of the participants.

	Categories	*n* (%)
Sociodemographic characteristics		
Marital status	Single	158 (45.1)
	Married/Common-law	188 (53.8)
	Other	4 (0.01)
Children	Yes	260 (74.3)
	No	90 (25.7)
Number of children	One child	92 (26.3)
	Two or more children	168 (48)
Level of education	Secondary school or less	138 (39.4)
	College/university/post-graduate	212 (60.6)
Employment	Employed	219 (62.6)
	Unemployed and Pensioner	131 (37.4)
Monthly income per family member	<40,000 ^a^ RSD	88 (25.1)
	40,000–60,000 RSD	228 (65.1)
	≥60,000 RSD	34 (9.8)
Health-related characteristics and health behaviors		
Chronic diseases	DMT2	189 (54)
	DMT2 and additional chronic diseases	161 (46)
HbA1c value	≤7%	216 (61.7)
	>7%	124 (35.4)
	No data	10 (2.9)
Therapy for DMT2	Diet and tablets	266 (76)
	Insulin/Insulin and tablets	84 (24)
Frequency of drug administration for DMT2	Once/Twice a day	224 (64)
	Three or more times a day	126 (36)
Active exercise	Never	57 (16.3)
	Less than once a week	135 (38.6)
	1–2 times a week	118 (33.7)
	3 or more times a week	40 (11.4)
Smoking	Smoker	178 (50.9)
	Nonsmoker	172 (49.1)
Alcohol	Never	156 (44.6)
	Once a month	121 (34.5)
	2 or more times a month	73 (20.9)
Access to health-related information and empowerment-related indicators		
Source of health information	Doctors	204 (58.3)
	Pharmacists	47 (13.4)
	Internet	15 (4.3)
	Other	84 (24)
Interest in health	Not interested/Little	142 (40.6)
	Medium	172 (49.1)
	Much and very interested	36 (10.3)
Self-estimation of health status	Bad	99 (28.3)
	Good	201 (57.4)
	Very good	50 (14.3)

^a^ 1 RSD = 0.0085 EUR.

**Table 2 ijerph-20-05190-t002:** Distribution of scores among the instruments.

	FHL	IHL	CHL	FCCHL-SR12	BRIEF-4	BRIEF-3
N	350	350	350	350	350	350
Mean	9.76	10.20	10.10	30.10	13.70	10.30
Median	10.00	10.00	10.00	30.00	14.00	10.00
SD	2.27	1.78	2.11	4.12	2.31	1.99
Minimum	4.00	5.00	4.00	16.00	8.00	6.00
Maximum	10.00	15.00	15.00	40.00	20.00	15.00
Skewness	0.31	0.07	0.09	0.25	0.20	0.40
Std. Error of Skewness	0.13	0.13	0.13	0.13	0.13	0.13
Kurtosis	1.00	0.09	0.10	0.91	−0.08	−0.09
Std. Error of Kurtosis	0.26	0.26	0.26	0.26	0.26	0.26

FHL—functional health literacy, IHL—communicative health literacy, and CHL—critical health literacy.

**Table 3 ijerph-20-05190-t003:** Observed variables associated with HL level by FCCHL-SR12, BRIEF-4, and BRIEF-3.

		FCCHL-SR12			BRIEF-4			BRIEF-3					
		InHL	MHL	AHL	InHL	MHL	AHL	InHL	MHL	AHL	FCCHL-SR12	BRIEF-4	BRIEF-3
		N (%)			N (%)			N (%)			χ^2^/*p*		
Gender	Male	57 (37)	97 (62)	2 (1)	65 (42)	77 (49)	14 (9)	79 (46)	72 (46)	13 (8)	4.83/0.089	10.08/0.006	14.04/0.001
	Female	59 (30)	125 (65)	10 (5)	53 (27)	108 (56)	33 (17)	58 (30)	97 (50)	39 (20)			
Marital status	Single	49 (31)	106 (67)	3 (2)	47 (30)	92 (58)	49 (12)	51 (32)	88 (56)	19 (12)	2.12/0.345	3.34/0.188	6.52/0.038
	Married/Common-law	64 (34)	116 (62)	8 (4)	70 (37)	91 (49)	27 (14)	37 (41)	79 (42)	32 (17)			
Children	No	26 (29)	60 (66)	5 (6)	28 (31)	53 (58)	10 (11)	29 (32)	51 (56)	11 (12)	2.54/0.638	12.20/0.016	19.81/0.001
	One child	32 (35)	57 (63)	2 (2)	20 (22)	54 (59)	17 (19)	20 (22)	54 (59)	17 (19)			
	Two or more children	58 (35)	105 (63)	5 (3)	70 (42)	78 (46)	20 (12)	80 (48)	64 (38)	24 (14)			
Level of education	Secondary school or less	17 (41)	78 (57)	3 (2)	60 (44)	68 (49)	10 (7)	67 (48)	59 (43)	12 (9)	7.33/0.025	13.48/0.001	15.71/<0.001
	College/university/post-graduate	59 (28)	114 (68)	9 (4)	58 (27)	117 (55)	37 (18)	62 (29)	110 (52)	40 (19)			
Employment	Employed	69 (31)	42 (65)	8 (4)	62 (27)	124 (57)	35 (16)	65 (30)	113 (52)	41 (19)	0.74/0.690	11.34/0.003	15.38/<0.001
	Unemployed or Pensioner	47 (36)	80 (61)	4 (3)	58 (44)	61 (47)	12 (9)	64 (49)	56 (43)	11 (8)			
Monthly income per family member	<40,000 RSD	31 (35)	54 (61)	3 (4)	29 (33)	52 (59)	7 (8)	39 (44)	40 (46)	9 (10)	9.87/0.043	11.67/0.020	3.75/0.440
	40,000–60,000 RSD	78 (34)	145 (65)	5 (2)	74 (33)	114 (50)	40 (17)	78 (34)	113 (50)	37 (16)			
	≥60,000 RSD	7 (21)	23 (68)	4 (12)	15 (44)	19 (56)	0 (0)	12 (35)	16 (47)	6 (18)			
HbA1c	≤7%	66 (31)	141 (65)	9 (4)	70 (32)	119 (55)	27 (13)	74 (34)	109 (51)	33 (15)	1.99/0.370	1.64/0.439	1.40/0.496
	>7%	46 (37)	75 (61)	3 (2)	44 (36)	60 (48)	20 (16)	50 (40)	55 (45)	19 (15)			
Therapy for DMT2	Diet and tablets	96 (36)	159 (60)	11 (4)	95 (36)	140 (153)	31 (11)	101 (38)	132 (50)	33 (12)	6.85/0.033	3.92/0.141	5.26/0.072
	Insulin/Insulin and tablets	20 (24)	63 (75)	1 (1)	23 (27)	45 (54)	16 (19)	28 (33)	37 (44)	19 (23)			
Frequency of drug administration	Once/Twice a day	83 (37)	132 (59)	9 (4)	78 (35)	119 (53)	27 (12)	85 (38)	107 (48)	32 (14)	5.48/0.064	1.11/0.574	0.37/0.830
	Three or more times a day	33 (26)	90 (71)	3 (3)	40 (32)	66 (52)	20 (16)	44 (35)	62 (49)	20 (16)			
Active exercise	Never	24 (42)	29 (51)	4 (7)	22 (39)	26 (46)	9 (16)	24 (42)	28 (49)	5 (9)	13.29/0.039	14.49/0.025	18.77/0.005
	Less than once a week	43 (32)	86 (64)	6 (4)	49 (36)	71 (53)	15 (11)	55 (41)	60 (44)	20 (15)			
	1–2 times a week	42 (36)	74 (63)	2 (2)	39 (33)	68 (58)	11 (9)	42 (36)	63 (53)	13 (11)			
	3 or more times a week	7 (18)	33 (82)	0 (0)	8 (20)	20 (50)	12 (30)	8 (20)	18 (45)	14 (35)			
Smoking	Smoker	69 (39)	104 (58)	5 (3)	67 (38)	92 (52)	19 (11)	75 (42)	85 (48)	18 (10)	5.28/0.071	3.79/0.150	8.24/0.016
	Non-smoker	47 (27)	118 (99)	7 (4)	51 (30)	93 (54)	28 (16)	54 (31)	84 (49)	39 (20)			
Alcohol	Never	44 (28)	105 (67)	7 (5)	48 (31)	79 (51)	29 (19)	53 (34)	68 (44)	35 (22)	11.75/0.019	11.02/0.026	18.28/0.001
	Once a month	36 (30)	82 (68)	3 (2)	43 (36)	62 (51)	16 (13)	50 (41)	56 (46)	15 (123)			
	2 or more times a month	36 (49)	35 (98)	2 (3)	7 (37)	44 (60)	2(3)	26 (35)	45 (62)	2 (3)			
Source of health information	Doctor	58 (28)	130 (68)	8 (4)	70 (34)	102 (50)	32 (16)	73 36)	94 (46)	37 (18)	6.61/0.358	11.81/0.066	17.83/0.007
	Pharmacists	21 (45)	25 (53)	1 (2)	13 (28)	24 (51)	10 (1)	14 (30)	22 (47)	11 (23)			
	Internet	6 (40)	8 (53)	1 (7)	3 (20)	11 (72)	1 (8)	3 (20)	11 (73)	1 (7)			
	Other	31 (37)	51 (61)	2 (2)	32 (38)	48 (57)	4 (5)	39 (46)	42 (50)	3 (4)			
Interest in health	Not interested/Little	50 (35)	88 (62)	4 (3)	48 (34)	81 (57)	13(9)	55 (39)	7 (54)	10 (7)	4.93/0.294	34.91/<0.001	33.45/<0.001
	Medium	59 (33)	106 (62)	8 (5)	59 (34)	95 (55)	18 (11)	62 (36)	84 (49)	26 (15)			
	Much or very interested	8 (22)	28 (78)	0 (0)	11 (31)	9 (25)	16 (64)	12 (33)	8 (23)	16 (44)			
Self-estimation of health status	Bad	28 (28)	64 (70)	2 (2)	30 (30)	53 (54)	16 (16)	38 (38)	50 (51)	11 (11)	2.66/0.616	8.53/0.074	14.56/0.006
	Good	70 (35)	123 (61)	8 (4)	72 (36)	99 (49)	30 (15)	77 (38)	85 (42)	39 (19)			
	Very good	18 (36)	30 (60)	2 (4)	16 (32)	33 (66)	1 (20)	14 (280)	34 (68)	2 (4)			

AHL—adequate health literacy, MHL—marginal health literacy, InHL—inadequate health literacy.

**Table 4 ijerph-20-05190-t004:** Sociodemographic characteristics as predictors for InHL.

Univariate Analyses	BRIEF-4	BRIEF-3	FCCHL-SR12
	OR (95% CI) *p* Value		
Age *	1.054 (1.028–1.079) 0.002	1.048 (1.024–1.073) *p* < 0.001	/
Gender (Female)	0.526 (0.336–824) *p* = 0.005	0.511 (0.329–0.793) *p* = 0.003	
Education (College/university/post-graduate)	0.700 (0.558–0.877) *p* = 0.002	0.662 (0.530–0.827) *p* < 0.001	0.740 (0.590–0.928) *p* = 0.009
Number of children *	1.356 (1.030–1.785) 0.030	1.516 (1.1154–1.993) *p* = 0.003	/
Employment (Unemployed or pensioner)	2.105 (1.336–3.319) 0.001	2.263 (1.445–3.545) *p* < 0.001	
Active exercise *	/	0.755 (0.589–0.968) *p* = 0.026	
Smoking (Smokers)	/	0.793 (0.637–0.987) *p* = 0.038	0.771 (0.615–0.966) *p* = 0.024
Alcohol *	/		1.518 (1.138–2.025) *p* = 0.004
Therapy for DMT2 (Insulin/Insulin and tablets)			0.553 (0.316–0.970) *p* = 0.039
Frequency of drug administration (Three or more times a day)			0.603 (0.373–0.975) *p* = 0.039
Multivariate analysis	BRIEF-4	BRIEF-3	FCCHL-SR12
	OR (95% CI) *p* value		
Age	1.057 (1.019–1.096) 0.003	1.042 (1.005–1.081) *p* = 0.025	/
Education (College/university/post-graduate)	0.733 (0.574–0.934) *p* = 0.012	0.727 (0.570–0.928) *p* = 0.010	0.766 (0.606–0.968) *p* = 0.026
Number of children *	/	1.403 (1.051–1.873) *p* = 0.022	/
Alcohol *	/	/	1.484 (1.10–2.00) *p* = 0.010

* Age—continuous variable; Number of children—ordered variable (no, one child, two or more children were coded from 1 to 3, respectively); Alcohol—ordered variable (never, once a month, 2 or more times a month were ordered from 1 to 3, respectively); Active exercise–ordered variable (never, less than once a week, 1–2 times a week, 3 or more times a week were coded from 1 to 4, respectively). Multivariate analyses were performed with significant predictors from univariate analyses.

## Data Availability

Data and the instruments could be available upon request.

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
