# Peer review of "Predictors of Inadequate Health Literacy among Patients with Type 2 Diabetes Mellitus: Assessment with Different Self-Reported Instruments"

_ijerph, 2023, doi:10.3390/ijerph20065190_

Round 1

Reviewer 1 Report

Background:

The introduction states the problem well and conducts the reader to the research question. The description of the questionnaires should be placed in the methods heading and not in the introduction.

Methods:

The exclusion criteria are established before the beginning of the study. You can not consider those who answered below 90% as excluded: they are non-responders and should be counted at least to know the response rate.

The final sample for analysis should be placed on the results and not at the methods.

The description of the questionnaires you used should be placed on a separate heading and should include a description of the psychometric properties of the original scale and the translation.

The STROBE checklist is a help for publishing, not for research design.

Results:

It misses the description of partial indexes of FCCHL questionnaire. You should explain your decision to compare the partial indexes with BRIEF 3 and 4 scores and with the whole score of FCCHL (figure 1 and table 2)

Acronyms of the tables or figures should be described in the respective subtitle.

Figure 2 refers to a classical proportion of concordance, but results in both the figure nor in the text aren’t presented like it.

Table 3 is very confusing. You try to include all the questionnaires in the same table, and the reader doesn’t understand what the associations you found are. For instance. You present a significant difference between gender in brief-3 and brief-4. I don’t understand if the difference is about the level of HL (Inadequate, Marginal or Adequate) or about gender itself. Also, it’s difficult to read the distribution according to gender. It seems that the proportions you found are related to the distribution of gender in the sample and not to the distribution of gender in the categories under analysis.

Also, table 4 needs some clarification. Is this a multivariate analysis? What are the variables for adjustment? Why didn’t you include variables with significant differences shown on table 3?

Discussion:

You must decide which scale better describe the HL in Serbian DMt2 patients and discuss your results from it, answering directly to your main objective. The discussion of the differences between the 3 questionnaires is crucial but secondary to this decision, allowing, for instance, to characterize the power of the test to describe HL in this population.

You present several results in the discussion that should be placed above in the text ([OR for parents with two children and OR for parents with more than two children).

Although brief 3 and 4 may serve as an additional confirmation, as you state, it is crucial that they point in the same way as FCCHL (I think you consider it the most accurate).

Although I tend to agree with the need for better education for health since basic school, I think your results don’t allow you to conclude that. Moreover, DMt2 is a disease of adults and children are too far from the problem to expect a great impact.

Abstract: should be corrected according the comments above

Author Response

Thank you for your valuable comments, please find attached our replies marked in yellow. Best regards

Reviewer 2 Report

Review for submission no. ijerph- 2184414

This is an interesting manuscript presenting findings from the study on predictors of inadequate health literacy (HL) among patients with type 2 diabetes mellitus. Such studies answer the rising burden of various chronic diseases and pave the way towards therapies tailored to individual patients’ needs.

The study itself seems well-designed and conducted. A comparative perspective (of different instruments to assess health literacy), as adopted by the authors, should lead to interesting and useful findings. However, the manuscript – in its current form – is not ready for publication.

Specific remarks:

1. This is to be a cross-sectional study (line 111). However, there is no explanation of the sample selection nor its representative character towards the Serbian population (in terms of variables such as education level, place of residence etc.). Explanation of the sample’s size computation is not enough to assess the study’s results as representative and meaningful for the population outside the city of Belgrade.

Authors should provide comprehensive information on the criteria used to select participants.

2. Manuscript presents a two-stage analysis of the results – firstly, of the demographic variables associated with HL level in each test (Table 3.), and secondly, of the characteristics that predict inadequate HL (using OR measure). The second part of the analysis (Table 4.) is limited to only four variables (age, education, number of children and alcohol), while there were many more variables identified as statistically meaningful for all or majority of HL tests (e.g. active exercise).

Authors should provide comprehensive information on the criteria used to select variables for which OR was calculated, as well as for the exclusion of other variables.

Moreover, detailed information is needed on what age, what amount of alcohol consumption and having of how many children result in inadequate HL (this refers also to lines 34-35).

3. This study aims population of patients living with type 2 DM.

However, the proposed discussion lacks a comparison of the results of HL assessments obtained in this and other studies that aimed at patients with other diagnoses (or DM diagnosis but living in other countries).

Authors should provide a more in-depth view to identify and present dis-/similarities between the results of their study (assuming its representative character) and other studies, as well as propose explanations to those findings.

4. Conclusion of the article should be rewritten as it fails to provide new information and does not correspond with a question on the predictors of inadequate HL among patients with type 2 DM.

5. The language of the manuscript is often unclear. Sentences are often too long and with too many subordinate clauses. Authors tend to use both passive voice and first-person plural narration, as well as different tenses throughout the text.

Author Response

(The authors gave the same response as above.)

Round 2

Reviewer 1 Report

No further comments.

Authors answered well and accordingly my previous comments.

Author Response

Thank you for your valuable comments.

Best regards! 

Reviewer 2 Report

I applaud the authors for all improvements to the manuscript.

In its current form, the paper provides a better explanation of materials and methods, thus allowing for a better undersending of the conclusions drawn.

1. However, new statements (e.g. in discussion paragraphs) require appropriate references. 

2. Some minor (grammar/spelling) mistakes should be addressed during final proofreading (after all editing is done).

Author Response

Thank you for your valuable comments. Please find attached updated manuscript and our comments. 

Best regards! 
